Ambulacrarian insulin-related peptides and their putative receptors suggest how insulin and similar peptides may have evolved from insulin-like growth factor

Veenstra Jan A. jan-adrianus.veenstra@u-bordeaux.fr
INCIA UMR 5287 CNRS, Université de Bordeaux , Pessac , Gironde , France
Holland Linda
Electronic publication date: 2021 Jul 14
Publication date: 2021
Volume: 9
Electronic Location ID: e11799
Received 2021 Apr 16; Accepted 2021 Jun 25
Copyright: ©2021 Veenstra
Copyright year: 2021
Copyright holder: Veenstra
License: This is an open access article distributed under the terms of the Creative Commons Attribution License, which permits unrestricted use, distribution, reproduction and adaptation in any medium and for any purpose provided that it is properly attributed. For attribution, the original author(s), title, publication source (PeerJ) and either DOI or URL of the article must be cited.
License URL: https://creativecommons.org/licenses/by/4.0/

Keywords: Insulin, IGF, Relaxin, Octinsulin, Multinsulin, Dilp7, GPCR, Gonadulin, Evolution, Receptor tyrosine kinase

Funding: CNRS The University of Bordeaux This work was supported by institutional funding by the CNRS and the University of Bordeaux. The funders had no role in study design, data collection and analysis, decision to publish, or preparation of the manuscript.

==============================
Background

Some insulin/IGF-related peptides (irps) stimulate a receptor tyrosine kinase (RTK) that transfers the extracellular hormonal signal into an intracellular response. Other irps, such as relaxin, do not use an RTK, but a G-protein coupled receptor (GPCR). This is unusual since evolutionarily related hormones typically either use the same or paralogous receptors. In arthropods three different irps, i.e. arthropod IGF, gonadulin and Drosophila insulin-like peptide 7 (dilp7), likely evolved from a gene triplication, as in several species genes encoding these three peptides are located next to one another on the same chromosomal fragment. These arthropod irps have homologs in vertebrates, suggesting that the initial gene triplication was perhaps already present in the last common ancestor of deuterostomes and protostomes. It would be interesting to know whether this is indeed so and how insulin might be related to this trio of irps.

Methodology

Genes encoding irps as well as their putative receptors were identified in genomes and transcriptomes from echinoderms and hemichordates.

Results

A similar triplet of genes coding for irps also occurs in some ambulacrarians. Two of these are orthologs of arthropod IGF and dilp7 and the third is likely a gonadulin ortholog. In echinoderms, two novel irps emerged, gonad stimulating substance (GSS) and multinsulin, likely from gene duplications of the IGF and dilp7-like genes respectively. The structures of GSS diverged considerably from IGF, which would suggest they use different receptors from IGF, but no novel irp receptors evolved. If IGF and GSS use different receptors, and the evolution of GSS from a gene duplication of IGF is not associated with the appearance of a novel receptor, while irps are known to use two different types of receptors, the ancestor of GSS and IGF might have acted on both types of receptors while one or both of its descendants act on only one. There are three ambulacrarian GPCRs that have amino acid sequences suggestive of being irp GPCRs, two of these are orthologs of the gonadulin and dilp7 receptors. This suggests that the third might be an IGF receptor, and that by deduction, GSS only acts on the RTK. The evolution of GSS from IGF may represent a pattern, where IGF gene duplications lead to novel genes coding for shorter peptides that activate an RTK. It is likely this is how insulin and the insect neuroendocrine irps evolved independently from IGF.

Conclusion

The local gene triplication described from arthropods that yielded three genes encoding irps was already present in the last common ancestor of protostomes and deuterostomes. It seems plausible that irps, such as those produced by neuroendocrine cells in the brain of insects and echinoderm GSS evolved independently from IGF and, thus, are not true orthologs, but the result of convergent evolution.

Introduction

Many protein hormone and neuropeptide signaling pathways have orthologs in both protostomes and deutertostomes showing that these pathways were already present in their last common bilaterian ancestor. In some cases, the orthologs of the peptide ligands show only limited sequence similarity, but their receptors contain protein domains that are sufficiently conserved to establish homology. Virtually all ligands employ either a single receptor or a number of related receptors that evolved by gene duplication. Co-evolution of peptide ligands and receptors insures that related protein hormones or neuropeptides use receptors akin to those of their orthologs (Mirabeau & Joly, 2013; Hsueh & Feng, 2020).

Insulin/IGF-related peptides (irps) are an exception to this rule. Whereas insulin and IGF act through a receptor tyrosine kinase (RTK), relaxin uses a leucine-rich repeat G-protein coupled receptor (LGR). This raises the interesting question as how this apparent jump from one type of receptor to another may have come about. In cockroaches, termites and stick insects three different irp genes (i.e., gonadulin, arthropod insulin-like growth factor (aIGF) and arthropod relaxin) are located next to one another in the genome and thus likely originated from a local gene triplication (Veenstra, 2020b). To avoid confusion with the vertebrate relaxins and related peptides, the arthropod relaxins will be referred to as Drosophila ilp7 (dilp7) in this manuscript. One of the irps, aIGF, is known to use an insulin RTK, while gonadulin acts through insect LGR3 (Vallejo et al., 2015; Garelli et al., 2015; Colombani et al., 2015). Bioinformatic evidence suggested that dilp7 must be the ligand for insect LGR4, and this has now been confirmed experimentally in Drosophila (Veenstra, Rombauts & Grbić, 2012; Imambocus et al., 2020), but dilp7 may also activate an RTK (Linneweber et al., 2014). This suggests that the archtype arthropod IGF-related peptide acted through both an RTK and an LGR and that after a likely gene triplication, some of the ligands may have lost one of the two original receptors. Although it is possible that the gene triplication of the ancestral insulin gene occurred in an early arthropod or protostomian, it may well have occurred in a bilaterian ancestor, as homologs of both aIGF and dilp7 are also present in deuterostomes.

Brain neuroendocrine insect irps are more closely related to IGF than either dilp7 or gonadulin. Therefore, a gene duplication that gave rise to separate genes encoding these peptides is likely to have occurred after the triplication that gave rise to the ancestor genes of gonadulin and dilp7. Yet in insect genomes irp genes are not located near the IGF gene. Thus, the particular organization of these genes suggests that whereas the gonadulin and dilp7 genes likely originated by two successive local gene duplications, the IGF gene duplication that gave rise to an initial arthropod neuroendocrine brain irp must have materialized in a different fashion. If the earlier mentioned gene triplication was already present in the last common ancestor of the deuterostomes then a similar argument can also be made for the evolution of insulin. Given the importance of insulin as a human hormone and the inherent interest of its evolutionary origin, I explored the evolution of bilaterian insulin-related peptides in more detail and here report on the genes coding for such peptides and their receptors in the Ambulacraria that suggest how insulin may have evolved from IGF.

Material and Methods

Nomenclature

Hormones have often been discovered independently by different groups using different bioassays. The vertebrate insulin-like growth factors are a good example of that. Predicted protostomian peptides and their receptors have sometimes been given names that refer to similar deuterostomian proteins. In some cases this is very confusing, e.g., vertebrate LGR-3, -4 and -5 are not the orthologs of arthropod receptors that have been given the same names. A similar problem occurs with arthropod relaxin that is not an ortholog of vertebrate relaxin. This peptide will therefore be called dilp7 (Drosophila insulin-like peptide 7). I will refer to arthropod LGR3 as the gonadulin receptor, arthropod LGR4 as the dilp7 receptor and arthropod LGR5 as GRL101, a GPCR initially identified from the pond snail Lymnaea stagnalis (Tensen et al., 1994) that is an ortholog of arthropod LGR5 (Veenstra, 2020b).

Another nomenclature problem concerns the terms, insulin-like and insulin-related that are not well defined. Insulin and IGF are related and must share a common evolutionary origin with other peptide ligands like vertebrate relaxin, INSL3, arthropod dilp7 and gonadulin and a large number of other bilaterian peptides. All these peptides are often collectively called insulin-like or insulin-related without any specification as to in which aspects these hormones are similar to insulin. The typical core sequence of six cysteine residues and its use of an RTK are two characters that are shared by vertebrate IGF and insulin. However, several related peptides have eight cysteine residues and others like vertebrate relaxin use an LGR and not an RTK. Insulin and IGF are also different in that IGF is a single chain molecule, while the insulin precursor is processed into a two chain molecule. The term insulin-like seems more appropriate for a subset of the insulin/IGF-related peptides that look similar to insulin and act through an RTK, yet are different from IGF. Calling IGF-related peptides, like vertebrate relaxin, INSL3 or arthropod gonadulin for which there is no evidence that they act through an RTK, insulin-like is confusing. Unfortunately for many bilaterian peptides we can only speculate as to which type of receptor they use. The difference between one or two chain ligands, i.e., IGF versus insulin, is also useless as there is good evidence that some insect IGF-related peptides are processed into two-chain molecules when expressed in neuroendocrine cells and produced as single chain ligands when produced by the fat body, yet in both cases stimulate an RTK. It is for these reasons that all these peptides will be referred to as insulin/IGF-related peptides, abbreviated irps.

Sequence analysis

Sequences for insulin related peptides and their likely receptors were identified from a number of Ambulacraria species. This was done using the Artemis program (Rutherford et al., 2000) and the BLAST+ program (https://blast.ncbi.nlm.nih.gov/Blast.cgi) on publicly available genome sequences from the feather star Anneissia japonica, the sea urchins Lytechinus variegatus (Davidson et al., 2020) and Strongylocentrus purpuratus (Sea Urchin Genome Sequencing Consortium, 2006), the sea cucumbers Apostichopus japonicus (Jo et al., 2017; Zhang et al., 2017) and Holothuria glaberrima, the sea stars Acanthaster planci (Hall et al., 2017), Pisaster ochraceus (Ruiz-Ramos et al., 2020) and Patiria miniata, the brittle star Ophiothrix spiculata and the hemichordates Saccoglossus kowalevskii and Ptychodera flava (Simakov et al., 2015). The genomes were downloaded from https://www.ncbi.nlm.nih.gov/genome. For many of these species there are also significant amounts of RNAseq data. These were analyzed using the sratoolkit (https://trace.ncbi.nlm.nih.gov/Traces/sra/sra.cgi?view=software) in combination with Trinity (Grabherr et al., 2011) by methods described in detail elsewhere (Veenstra, 2020b). Some protein sequences were found in the NCBI database, but several of them contain errors or are incomplete. Where possible these were corrected and/or completed using the methods described above. As there is only a single crinoid genome assembly available, transcriptome data from the three crinoid species Antedon mediterranea, Florometra serratissima and Oligometra serripinna were also included. For the same reason transcriptome data from the brittle star Amphiura filiformis, Ophioderma brevispina and the hemichordate Schizocardium californicum were likewise analyzed. Obviously, transcriptome data can only demonstrate the presence of gene but not its absence, and their usefulness depends largely on the variety of tissues sampled and the expression levels of the genes of interest. Nevertheless, such data often provide additional sequences that, even if they are incomplete, increase the robustness of sequence comparisons. Genomic and transcriptomic RNAseq short read archives (SRAs) were downloaded from NCBI (https://www.ncbi.nlm.nih.gov/sra/); a list of the SRAs analyzed is provided in the supplementary data.

As queries for the insulin-like peptides, a number of such peptides from a variety of species were used. Insulin RTKs are easily identified in genome and transcriptome assemblies, as their kinase domains are very well conserved. The LGRs that could function as insulin receptors are more variable. Vertebrate LGRs RXFP1 and RXFP2 are known receptors for relaxin and Ins3 and Drosophila LGR3 and LGR4 for gonadulin, and dilp7 respectively. Other LGRs function as receptors for the various glycoprotein hormones, GPA2/GPB5, bursicon, TSH, FSH and LH. These GPCRs cluster on phylogenetic trees with another protostomian LGR, GRL101. This GPCR was initially identified from the pond snail Lymnaea stagnalis and was the first GPCR discovered to have, in addition to six leucine-rich repeats, twelve repeats of a sequence that was known to exist in the low density lipoprotein receptor and are now called LDLa repeats (Tensen et al., 1994). I have previously suggested (Veenstra, 2020b) that this receptor might be an IGF receptor.

Both the RTK and LGR receptors have large ectodomains. Those of the insulin RTKs are very similar from one receptor to another, while those of the LGRs differ between different types. The latter all contain numerous Leucine-rich repeats (LRRs), and some also have LDL-receptor class A (LDLa) repeats. Both LRRs and LDLas are present in many other proteins. Initial searches for orthologous receptors were, therefore, done using the transmembrane regions of various insect and vertebrate LGRs and the protein kinase domain of RTK. Once partial sequences of putative receptors were identified, the coding sequences of these domains were then used to complete the cDNA sequences as best as possible, using either Trinity on RNAseq SRAs or Artemis on genome sequences.

Sequence similarity and phylogenetic trees

Both phylogenetic and sequence similarity trees use Clustal omega (Sievers et al., 2011) to produce alignments. Fasttree (Price, Dehal & Arkin, 2010), using the ./FastTreeDbl command with the -spr 4, -mlacc 2 and -slownni options, was used to construct trees and estimate probabilities.

In order to identify putative receptors for the various irps, LGRs that show homology to various arthropod and vertebrate LGRs were identified and a phylogenetic tree based exclusively on the transmembrane regions of these receptors was constructed.

Precursor processing

Precursors of insulin-like peptides contain signal peptides that are removed on entry into the endoplasmatic reticulum. Signal P 5.0 (Almagro Armenteros et al., 2019) was used online (http://www.cbs.dtu.dk/services/SignalP/) to predict where this cleavage would most likely occur. Some, but not all, precursors are further processed by convertases. Of these furin is ubiquitously present in all cell types and can thus potentially cleave any secreted protein with an appropriate cleavage site. Its consensus cleavage site is K/R-X-K/R-R; the two human IGF precursors are processed at KSAR and KSER, respectively (Humbel, 1990). Precursors that are produced in cells with a regulated pathway, such as neuroendocrine and enteroendocrine cells, are also exposed to other convertases like PC1/3 and PC2. Their consensus cleavages site is KR. However, effective proteolytic processing by convertases is strongly influenced by amino acid residues surrounding these consensus cleavage sites. For example, bulky residues immediately following the arginine residue, a proline residue before the consensus site, or disulfide bridges nearby can cause sufficient steric hindrance to inhibit cleavage. Using rules proposed to predict cleavage by PC1/3 and PC2 in both vertebrates and insects (Devi, 1991; Rholam et al., 1995; Veenstra, 2000), I have tried to indicate where the various precursors might be cleaved. It must be noted, however, that there is no certainty that these sites will be cleaved, nor can it be excluded that proteolytic processing occurs at sites that have not been indicated as such.

Expression

With a few notable exceptions (e.g., Lin et al., 2017), little is known about the expression of the various insulin-like peptides in either echinoderms or hemichordates. Except for the GSS our knowledge of their functions is also very limited. Expression data may reveal some preliminary clues as to where and when they are expressed and thus provide a hint as to their function. For this reason the number of reads corresponding to the various insulin-related peptides and their putative receptors was determined in a number of SRAs to provide evidence as to the time and tissue specific expression of these proteins. The analysis was performed as described previously (Veenstra, 2020b), and the data are supplied in Spreadsheet S2.

Results

Peptides related to insulin and IGF

Some protein sequences were found in the NCBI database, but several of them contain errors or are incomplete. Where possible, these were corrected and/or completed using the methods described above. As there is only a single crinoid genome assembly available, transcriptome data from the three crinoid species Antedon mediterranea, Florometra serratissima and Oligometra serripinna were also included. For the same reason transcriptome data from the brittle star Amphiura filliformis, Ophioderma brevispina and the hemichordate Schizocardium californicum were likewise analyzed. Obviously, transcriptome data can only demonstrate the presence of a gene but not its absence and their usefulness depends largely on the variety of tissues sampled and the expression levels of the genes of interest. Nevertheless, such data often provide additional sequences that even if they are incomplete, increase the robustness of sequence comparisons.

Insulin-like peptide precursors are typically characterized as having A, B and C domains that correspond to the A- and B-chains of insulin and the connecting peptide respectively. In IGF, D and E domains are also recognized, in which the D domain refers to the extension of the A chain and the E domain to part of the precursor after the D domain that is cleaved from IGF in the Golgi apparatus. For dilp7 orthologs it is appropriate to add an F (front) domain for the sequence in the N-terminal of the B-chain that in some peptides is not only larger, but also well conserved (Fig. 1).

Figure 1 Domains of insulin/IGF-related peptides.

Human insulin and IGF and Drosophila dilp7 are aligned and the different domains that are recognized in the precursors of these peptides are indicated. In insulin the domain borders are the convertase cleavage sites that are hihglighted in red. The A- and B-domains of insulin correspond to the A- and B-chains of insulina and the C-domain to the connecting peptide. Although IGF consists of a single protein chain due to its strong sequences similarity to insulin the A- and B-domains correspond to homologous regions of those domains in insulin, while the C-domain is the sequence between the A- and B-domains. In insulin there is only a single amino acid residues after the last cysteine residue, but in IGF there is a longer sequence, that has been called the D-domain. The IGF precursor is cleaved by furin in the Golgi apparatus and the sequence that is removed has been called the E-domain. Dilp7 is only known from nucleotide sequences, it is unknown how the precursor is exactly processed. Nevertheless, the presence of putative convertase cleavage sites, highlighted in red, suggests the presence of A-, B- and C-domains quite similar to those in insulin. However, unlike insulin or IGF, the putative B-chain of dilp7 has a long N-terminal extension that I propose to call the F-domain. The latter is well conserved in dilp7 orthologs from other bilaterians (Fig. 4).

Figure 2 Sequences of selected ambulacrarian IGF.

Partial IGF sequences from selected ambulacrarians are illustrated to show their sequence similarity. The A-, B- and C-domains of the insulin core are aligned, but not the putative D- and E- domains, as their amino acid sequence is only conserved in closely related species (Fig. S1). Not aligning D- and E- domains allows the visualization the context of putative convertase cleavage sites. None of the arginine or lysine residues conform to a typical arthropod or vertebrate convertase cleavage site. Although the sequence of the latter part of the IGF precursors is not well conserved, all of them are rich in positively charged amino acid residues. Conserved cysteine residues are indicated in red, conserved amino acid residues are highlighted in black and conserved substitutions in grey. The arginine and lysine residues in the D- and E- domains are highlighted in blue.

Previous work on insulin-related peptides in echinoderms have identified two different types of insulin-like peptides, gonad-stimulating substances (GSS) and insulin-like growth factors (Mita et al., 2009; Perillo & Arnone, 2014; Semmens et al., 2016; Smith et al., 2019). The insulin-like growth factors, but not GSS, are also present in hemichordates. While only a single IGF gene was found in the crinoids and hemichordates, other ambulacrarians have two such genes (Fig. 2, Figs. S1, S2; Spreadsheet S1). These proteins have large C-terminal extensions that are rich in charged amino acid residues, especially arginine and lysine, but also aspartic and glutamic acid residues. A comparison of the protein sequences and cDNAs from human IGFs identifies the exact separation between the D and E domains in these proteins (Humbel, 1990). However, although the corresponding sequences of the hemichordate and echinoderm IGFs contain numerous arginine and lysine residues (Fig. 2, Figs. S1, S2), there are no obvious convertase cleavage sites as many potential arginine residues are succeeded by residues known to inhibit such enzymes in vertebrates. It is thus not impossible that the D domains of these proteins are much larger than in the vertebrate IGFs and, if so, likely contain numerous positively charged amino acid residues. There are few transcriptome SRAs for specific tissues, however, the data that is available, suggest that the IGFs are expressed by many tissues, with the ovary showing significant expression. Patiria pectinifera is the only species with follicle cell specific SRAs and IGF-1 is strongly expressed by these cells and its mRNA is probably transferred to the oocyte (Spreadsheet S1).

The GSS are known to induce oocyte maturation and ovulation in a two step process, where GSS stimulates the follicle cells to produce 1-methyladenine which subsequently induces resumption of meiosis in the oocyte and about 30 min later this is followed by ovulation (Chiba, 2020). Interestingly, GSS was not found in either the genome nor the extensive transcriptome data from the feather star Anneissia japonica and was similarly not encountered in the transcriptomes of three other crinoids (Data S1). Transcriptomes may miss the expression of some genes, and large genome assemblies are never perfect. The short sequence reads in the genomic SRAs from Anneissia were therefore also analyzed for the presence of GSS, but again no evidence for such a gene was found. This peptide is thus likely absent from Anneissia and perhaps all Crinoidea. In the Holothuroidea and the Asterozoa, but not the Echinoidea, this gene is duplicated with the two paralogous peptides showing significant sequence variability (Fig. 3, Figs. S3, S4; Spreadsheet S1). As for all these peptides and their putative receptors, expression data are very limited, but in Apostichopus the two GSSs are differentially expressed, with GSS-1 being expressed at specific stages during embryonic development as well as by muscle and GSS-2 strongly expressed by both the ovary and the testes. Interestingly, in Holothuria scabra, it is the ortholog of GSS-1 that has been tested for biological activity and induces ovulation (Chieu et al., 2019). This makes one wonder what the effects of GSS-2 on ovulation might be in this species. However, Apostichopus was the only species where a significant GSS expression was found in the gonads (Spreadsheet S2).

Figure 3 Sequences of selected echinoderm GSS.

Sequence alignment of a few echinoderm GSS showing relatively conserved A- and B- domains of the insulin core sequence and likely KR convertase cleavage sites that can be expected to be cleaved by neuroendocrine convertase as well as a few potential furin sites. Conserved cysteine residues are indicated in red, conserved amino acid residues are highlighted in black and conserved substitutions in grey. The arginine and lysine residues that form likely –or possibly in the case of Apostichopus GSS-2 - part of a convertase site are highlighted in blue. For the alignment of a larger number or echinoderm GSS sequences see Fig. S3.

Figure 4 Sequences of selected dilp7 orthologs.

Sequences of Drosophila dilp7 and several ambulacrarian orthologs illustrating well conserved sequences, not only in typical insulin core of the peptides, but also in the F-domain (underlined in blue). Note that the sequence conservation of these peptides is stronger than in the IGFs or GSSs (Figs. 1 and 2). Conserved cysteine residues are indicated in red, conserved amino acid residues are highlighted in black and conserved substitutions in grey. Likely convertase cleavage sites have been highlighted in blue. Sequences are from Spreadsheet S1 and (Veenstra, 2020b), a comparison of a larger number of sequences is presented in Figs. S5 and S6.

Two other insulin-like peptides are commonly present in both hemichordates and echinoderms, including the Crinoidea. The first is an ortholog dilp7, which has a very characteristic F domain while its A chain is also remarkably well conserved (Fig. 4, Figs. S5, S6; Spreadsheet S1). The precursors of this peptide contain typical neuroendocrine KR convertase sites and seem to have their highest expression in the nervous system, although expression also occurs in other tissues. During embryogenesis dilp7 expression occurs relatively late (Spreadsheet S2). The second peptide present in all ambulacrarians has been called octinsulin as it has eight cysteine residues and is thus predicted to have four rather than three disulfide bridges. In echinoderms octinsulin is a single copy gene, but hemichordates have several such genes (Fig. 5, Figs. S7, S8; Spreadsheet S1). Octinsulin expression levels are the highest in nervous tissue, and significant expression is also found in the gut and stomach of Strongylocentrotus and Patiria pectinifera respectively. Although virtually absent from normal gut in Apostichopus, it has significant expression during gut regeneration of this species (Spreadsheet S2).

Figure 5 Sequences of selected ambulacrarian octinsulins.

Sequence alignment of a number of octinsulin sequences show that these sequences all have typical neuroendocrine convertase KR cleavage sites, suggesting these precursors are processed by enteroendocrine and/or neuroendocrine cells. Conserved cysteine residues are indicated in red, conserved amino acid residues are highlighted in black and conserved substitutions in grey. Likely convertase cleavage sites have been highlighted in blue. Sequences are from Spreadsheet S1, a comparison of a larger number of sequences is presented in Figs. S7 and S8.

The Asterozoa have genes coding for a fifth type of insulin, which is usually present in multiple copies termed multinsulins. The predicted peptides share structural similarity with the dilp7 orthologs; their genes have typically four coding exons rather than the two or three of the other irp genes. The sprawl of these peptides is perhaps best illustrated by a phylogenetic tree that suggests independent multiplication of these genes in several species (Fig. S10). Within a single species the various multinsulins, thus, often seem more closely related to one another than to their putative orthologs in other Asterozoa. Some of the multinsulins, like the octinsulins, have acquired two additional cysteine residues and are thus predicted to have four disulfide bridges, but the location of these additional cysteine residues differs from that in octinsulins (Figs. 6 and 7, Figs. S9, S10; Spreadsheet S1). Like dilp7, the multinsulins have typical neuroendocrine KR convertase cleavage sites and can thus be expected to be expressed in neuroendocrine and/or enteroendocrine cells;however, expression data on P. pectinifera suggest a relatively ubiquitous expression in several tissues.

Figure 6 Sequence comparison of selected ambulacrarian multinsulins and dilp7 orthologs.

Three different sets of sequences are compared. The top five sequences are dilp7 orthologs, the next five are multinsulins having three disulfide bridges and the last five multinsulins having four disulfide bridges. Note that although the multinsulins and the dilp7 orthologs share some sequences similarity this does not include the F-domain. Like the octinsulins these sequences all have typical neuroendocrine convertase KR cleavage sites, suggesting they are processed by enteroendocrine and/or neuroendocrine cells. Conserved cysteine residues are indicated in red, conserved amino acid residues are highlighted in black and conserved substitutions in grey. Likely convertase cleavage sites have been highlighted in blue. Sequences are from Spreadsheet S1, a comparison of a larger number of sequences is presented in Figs. S9 and S10.

Figure 7 Position of introns in ambulacrarian irp genes.

Schematic representation of the location of the cysteine residues, indicated as purple rectangles, and introns, represented by green Ts, in the coding sequences of the various types of ambulacrarian insulin-like genes. Numbers indicate the phase of each intron. All genes share the typical phase 1 intron present in insulin-like genes, whereas dilp7 and multinsulin genes also share a phase 2 intron. Signal peptides indicated as interrupted bars.

The genome assemblies of A. planci and Pisaster ochraceus show these genes to be clustered in the genome and some RNAseq sequences suggests that at least on occasion coding exons from different genes may be combined (Fig. S10). This and the large numbers of SNPs typically present in animals caught in nature and used for RNAseq preparation make it impossible to reliably determine their exact numbers.

Genome assemblies allow identification of the introns in these genes. All insulin genes have a characteristic phase 1 intron somewhere in their conceptual C domain. This is the only intron in the coding sequences of the octinsulin and GSS genes. The IGF genes have a phase 0 intron near the end of the coding sequence, and at least some of them have another phase 1 intron just after the transcription start site. The genes coding for the dilp7 orthologs and multinsulins share an additional phase 2 intron, and the multinsulin genes have yet another phase 1 intron. All these introns appear perfectly conserved (Fig. 7).

Synteny of genes producing insulin-like peptides

In the Strongylocentrotus genome, all five genes are located on the same chromosome, with the two IGF genes and those encoding octinsulin and dilp7 orthologs next to one another, and GSS at a distance of 6,000,000 bp (base pairs). At least the Anneissia octinsulin and IGF genes are likely located next to one another on the same chromosome also, as in the current genome assembly two of the three coding exons of IGF and one of the two octinsulin coding exons are located within about 10,000 bp. The three missing exons of these two genes are all located on minicontigs of less than 2,000 bp, as is one of the coding exons for the dilp7 ortholog. The contigs of the Lytechinus variegatus genome assembly are smaller and this may explain why in this species the genes are located on three different scaffolds, with the two ILGF-like peptides and the octinsulin together on a single contig. However in the recently published genome of the closely related L. pictus (Warner et al., 2021) the dilp7 ortholog is also closely associated with the other three genes. The GSS gene is on the same chromosome but at a distance of 28,000,000 bp. In the Apostichus japonicus genome assembly the genes encoding the octinsulin and the two IGF genes are located on the same contig, and the other genes each on a different one. In the draft Holothuria glaberrima genome assembly only the two IGF genes are located on the same contig, however in a single Oxford nanopore read (SRR9125585.2851.1) from H. scabra the octinsulin, dilp7 and two IGF genes are located next to one another as well (Fig. 8).

Figure 8 Synteny of ambulacrarian irp genes.

Schematic representation of the relative localization of different irp genes in several arthropod and ambulacrarian genomes. Arrow heads indicate transcription direction of the various genes, the numbers below the line indicate the number of nucleotides between the coding regions of adjacent genes in kilo base pairs. Note that the relative organization of the two insects –the cockroach Blattella germanica and the stick insect Timema crisitinae –is the same as in the hemichordate Saccoglossus kovalewskii and remarkably similar to that of the sea urchin Strongylocentrotus purpuratus and the sea cucumber Holothuria scabra. In the spider Pardosa pseudoannulata and the sea cucumber Apostichopus japonicus some of the genes are also next to one another. However, in the sea stars Acanthaster planci and Pisaster ochraceus synteny has been lost. Arthropod data from (Veenstra, 2020b).

Whereas the various Echinozoa genome assemblies suggest a certain degree of synteny with regard to the various irp genes, the Asterozoa genomes show that such syntenty is disintegrating. This is most clearly demonstrated in the genome assemblies from Pisaster ochraceus and Acanthaster planci, where the scaffolds are much larger than from Patiria miniata. In these species synteny is largely lost (Fig. 8). Interestingly the various multinsulin genes are present in small clusters on different chromosomes in those species.

Sequence similarity tree peptides related to insulin

Peptides having the characteristic insulin signature are notoriously variable in their primary amino acid sequences. Although the various residues allow one to align those sequences, such alignments will not always yield reliable phylogenetic trees as the basic tenet of such analyses is often not met. As an alternative I have proposed to use “sequence similarity trees”. Such trees are constructed using the same methods but do not pretend to illustrate phylogenetic relations, rather similarities between the different proteins.

As the structures of the multinsulins are most similar to the dilp7 orthologs (Fig. 6), it is not surprising that the sequence similarity tree (Fig. 9) groups the multinsulins with the dilp7 orthologs. The hypothesis that this structural similarity between these two types of peptides may reflect a close evolutionary relationship is reinforced by the presence of an intron in the genes encoding these peptides but not in the genes encoding octinsulin, IGF and GSS (Fig. 7). The tree also illustrates significant sequence similarity between GSS and the IGF.

Figure 9 Radial sequence similarity tree of ambulacrarian irps.

Note that the GSSs are similar to IGFs and seem to be related to them, while the multinsulins are most similar to the dilp7 orthologs. Echinoderm branches are in black, hemichordate branches in red. More extensive sequence comparisons and sequence trees are in the supplementary data (Figs. S1–S10). All sequences are from Spreadsheet S1.

Orthologs of receptors for irps: receptor tyrosine kinase

A single insulin RTK gene was found in all species analyzed here. An alternatively spliced form is present in Acanthaster and is likely commonly present in echinoderms (Spreadsheet S1). Hundreds of ambulacrarian protein sequences were identified at NCBI using a BLAST search with the S. kowalevskii protein kinase domain as a query. After aligning them with Clustal omega the protein kinase domains were used to make a phylogenetic tree. Results revealed no other known or predicted proteins with a similar protein kinase domain. The insulin RTK is ubiquitously expressed (Spreadsheet S2).

Figure 10 Phylogenetic tree of LGRs.

Phylogenetic tree constructed from the transmembrane regions of ambulacrarian LGRs that are putative receptors for irps. A few human and insect sequences have been added for comparison. The insert at the top shows the same data to which the glycoprotein LGRs have been added and where characteristic ligands for each branch have been identified. Numbers in blue indicate the apparent probabilities as determined by Fasttree. For details of the glycoprotein LGRs see Fig. S11.

Orthologs of receptors for irps peptides: LGRs

LGR sequences were obtained using the combination of genomic sequences and, where available, transcriptome shotgun sequences and RNAseq SRAs. The latter were used to produce contigs using Trinity (Spreadsheet S1). Short read assemblers are good in combining sequences into larger continuous ones, but they do produce artifacts, which are more easily obtained when very similar sequences are present in multiple copies, such as the multinsulins, or the numerous LDLa and LRR repeats. These repeats are usually individually coded by single exons that are sometimes skipped, and when such skipped individual reads enter in the RNAseq SRA, incorrect constructs are obtained. Furthermore, these repeats are present in numerous proteins, and from time to time this leads to assembled sequences that are from mRNA species from different genes. It is therefore to be expected that not all assembled transcripts, neither those in the databank nor those produced here, will be correct. Some errors were corrected by challenging divergent sequences that were discovered on comparing putative orthologs with one another. Other differences could be confirmed as true differences, but it is not impossible that some errors remain, particularly for those sequences that are incomplete. LGRs that might function as receptors for the various irps were identified by their homology with such receptors from vertebrates and arthropods. The transmembrane regions of the GPCRs don’t have the assembly problems of the LDLa and LRR repeats and are the most characteristic domain of the GPCRs. This makes it easier to construct a phylogenetic trees for these receptors based on their transmembrane regions than that it is to produce complete LGR transcripts.

Results show a surprisingly similar distribution of LGRs in the species studied. The tree resolves two major branches, one for the glycoprotein hormone receptors, which itself is divided in two subbranches, one for orthologs of the GPA2/GPB5 receptor - containing the receptors for human TSH, FSH and LH - and a second one for the bursicon receptor orthologs. All species studied are represented by one member on each of these two subbranches, except for Ophiothrix, where the draft genome reveals two orthologs each for the bursicon and GPA2/GPB5 receptors (Fig. 10). These are likely receptors for the bursicon and GPA2/GPB5 orthologs identified from various echinoderm species (Semmens et al., 2016). It is interesting to see that whereas vertebrates have different receptors for TSH, FSH and LH, most echinoderms have only one GPA2/GPB5-receptor ortholog (Fig. S11), even though A. rubens has two GPA2 and three GPB5 orthologs (Semmens et al., 2016). The LGRs for the glycoproteins were included in the search for putative receptors for the ambulacrarian irp LGRs in order to be sure that no such receptors would be missed.

The lower branches of the LGR phylogenetic tree are the ones of interest as they contain receptors with irp ligands. It consists of three subbranches, that are characterized by Drosophila LGR3 and LGR4 –the receptors for gonadulin and dilp7 respectively - and Periplaneta LGR5, an ortholog of Lymnaea GRL101. Here in all ambulacrarian species studied only one ortholog was found for each of them, despite extensive attempts to find additional LGRs in the various genomes and transcriptomes.

The GRL101 transmembrane regions puts it very close to vertebrate glycoprotein hormone and relaxin LGRs. LRRs are present in many different proteins, but when the LRR part of the Anneissia GRL101 (amino acid residues 576-717) is used as query in a protein BLAST against human proteins, the glycoprotein hormone and relaxin receptors are identified as most similar to this ectodomain of GRL101, suggesting that similarity of the GRL101 receptors with vertebrate LGRs is not limited to the transmembrane region of this GPCR.

Sequence alignments of these GPCRs show strong sequence similarity (Figs. S12–S14), however the dilp7 receptor ortholog varies more between species. A schematic representation of the the ectodomains of the LGRs on this second branch is drawn in Fig. 11. The orthologs of the dilp7 and gonadulin receptors each have a single LDLa repeat, except for the Patiria and Acanthaster orthologs of the dilp7 receptor which both have two LDLa repeats (Fig. S13). This additional LDLa is likely due to a relatively recent duplication of the LDLa since the two LDLa repeats have very similar amino acid sequences (Spreadsheet S1). All three receptors are expressed in the nervous system and the gonadulin receptor is well expressed in the gonads, both testis and ovary, and strongly so in the follicle cells of P. pectinifera (Spreadsheet S2).

Figure 11 Ectodomains of ambulacrarian LGRs.

Schematic representation of the various domains of the putative receptors for ambulacrarian insulin-related peptides. Each green circle symbolizes an LDLa repeat and each purple rectangle an LRR repeat, while the yellow oval indicates the seven transmembrane regions. The top representation corresponds to the gonadulin and dilp7 receptors (Figs. S11, S12). Note though, that the latter are somewhat variable, notably in the sea stars of two species of the Patiria genus and Acanthaster planci those receptors have two LDLa repeats (for details see Fig. S12). The bottom representation corresponds to the GRL101 receptors (Fig. S13).

Discussion

The genomic and transcriptomic data from both the hemichordates and the echinoderms show that these two groups share three irps (octinsulin, IGF and a dilp7 orthologs). IGF and dilp7 are orthologs of the arthropod peptides that together with gonadulin originated from a gene triplication. The structure of gonadulin is poorly maintained, even within insects (Veenstra, 2020b). The variable structure of gonadulin and its loss in many arthropod lineages suggests that the evolutionary pressure on gonadulin is weak. This may explain why the amino acid sequence of gonadulin looks significantly different from octinsulin. Nevertheless, there are two lines of evidence that suggest that these peptides must be orthologs as well. For one, synteny of the chromosome fragment containing these genes is conserved between the sea urchin Strongylocentrotus purpuratus, the hemichordate Saccoglossus kowalevskii and the cockroach Blattella germanica, suggesting that these peptides are likely orthologs. More importantly, all ambulacrarians have an ortholog of the gonadulin receptor and the only plausible ligand for such a receptor encoded by their genomes is octinsulin. Thus, the gene triplication previously reported from arthropods must have occurred in a common bilaterian ancestor of the deuterostomes and protostomes.

Crinoids have the simplest irp signaling system, one gene each for IGF, octinsulin and the dilp7 ortholog. Their putative receptors - insulin RTK, GRL101, and the orthologs of the dilp7 and gonadulin receptors –similarly are also each coded by a single gene. The hemichordates have a very similar repertoire, except that the octinsulin gene is systematically amplified and in some species the dilp7 ortholog as well. It thus appears likely that the first deuterostome had a single copy of each of these genes.

Within the echinoderms, the irp genes evolved considerably, as shown both by an increase in their numbers and the loss of synteny. Whereas the feather stars appear to have only a single IGF gene, all other echinoderms have two such genes and two novel irps, GSS and multinsulin, appeared. The GSS sequences are most similar to those of IGF, suggesting that they evolved from a gene duplication event from the IGF gene. Although some GSS genes are located on the same chromosome as the other irps, they are not close to the IGF genes, indicating that the IGF-GSS split was not a local duplication but may have been the result of an incorrectly repaired chromosome break.

In the Asterozoa a fifth type of irp gene emerged, those that code for the multinsulins which share significant sequence similarity with the dilp7 orthologs. The initial multinsulin gene must thus have its origin in a gene duplication of the dilp7 ortholog gene, with which they also share a characteristic intron. Later the multinsulin gene seems to have undergone several additional gene duplications; in this respect the multinsulins resemble the insect neuroendocrine irps.

The co-evolution of ligands and receptors allows one to assign the putative receptors for gonadulin, the dilp7 ortholog and IGF as the orthologs of the receptors of their arthropod orthologs. This allows the identification of the ambulacrarian LGRs that are the orthologs of the gonadulin and dilp7 receptors as likely receptors for octinsulin and the dilp7 respectively, as well as the insulin RTK as a receptor for IGF.

The appearance of the multinsulins is not accompanied by the evolution of a novel insulin-receptor. Some animals have multiple insulin RTKs, e.g., some arthropods have up to four such genes (Veenstra, 2020a; Veenstra, 2020b); however, in spite of extensive searches for a second insulin RTK in ambulacrarian genomes, none was found. Searches for an additional LGR that might function as a receptor for the GSS and/or multinsulin were unsuccessful and this raises the question which receptors are activated by these peptides.

I have previously argued that the close chromosomal association of the IGF, gonadulin and dilp7 ortholog genes in basal insects suggests that they derived from a gene triplication (Veenstra, 2020b). There are different scenarios that can explain how IGF and gonadulin came to respectively activate an RTK and an LGR. It is possible that the original irp activated an RTK and that an LGR was later acquired as a second receptor by gonadulin, alternatively the original irp activated an LGR and IGF acquired an RTK as a second receptor. Given the importance of insulin RTKs for growth in very basal metazoans, it is improbable that the original irp activated an LGR and that an RTK was acquired much later during evolution (see e.g., Mortzfeld et al., 2019). This indicates that an irp acquired an LGR as a second receptor and the question is whether this happened before or after the gene triplication that yielded IGF, gonadulin and dilp7. In both Saccoglossus and arthropods the IGF gene is in the middle of the three. This suggests that this represents the gene organization after the gene triplication and that the dilp7 and gonadulin orthologs each evolved independently from the original irp rather than that dilp7 evolved from gonadulin or vice versa. Had dilp7 originated from a gene duplication of gonadulin or the other way round, they also might have been more similar to one another than they are. The acquisition of a second receptor must be an extremely rare event. Since both gonadulin and dilp7 use an LGR this would mean that such an extremely rare event of the acquisition of a second receptor would have occurred not only twice, but even with a very similar receptor. Furthermore, some metazoans have an LGR that is closely related to the dilp7 and gonadulin LGRs suggesting that it could be an IGF receptor (see below). It is for these reasons that the author favors the hypothesis that the original irp already acted on both an LRG and an RTK, but, clearly, this remains a hypothesis.

The binding of insulin and relaxin to their respective receptors has been resolved in much detail in the last couple of years. The effective binding and stimulation of insulin RTK by the small irp from the snail Conus to the RTK shows that a small irp can be an effective ligand for this receptor (Menting et al., 2015). On the other hand, the complex interaction of relaxin to its LGR makes it more difficult to imagine a smaller peptide as an effective ligand (Hoare et al., 2019). Furthermore, considering the well conserved F-domain of the dilp7 receptor orthologs it is likely that it is necessary for interaction with its LGR receptor. The loss of this structure in multinsulin suggests that it is unlikely to be a dilp7 receptor agonist. On the other hand, the poor sequence conservation in the various Drosophila irps that activate a single RTK is reminiscent of the large structural variability of the multinsulins. This seems to suggest that the multinsulins are RTK ligands rather than that they activate the LGR.

The emergence of the GSS is not accompanied by the evolution of a novel receptor for these irps. This can be explained by assuming that IGF acts on both the RTK and an LGR and that the GSS have lost their affinity for the LGR. This raises the question whether an IGF LGR might exist.

If there were an IGF LGR, one would expect it to be related to the gonadulin and dilp7 receptors. GRL101 appears to be a plausible candidate as its transmembrane regions are closely related to the receptors for gonadulin and dilp7. The ectodomain of GRL101 consists of two parts, a series of LRRs and a second series of LDLa’s. In the related GPCRs, the LRRs are expected to bind with the insulin core of gonadulin and dilp7 orthologs, just like the human relaxin receptors (Hoare et al., 2019). When the LRR part of the Anneissia GRL101, the most basal echinoderm, was used as query for similar human proteins in a BLAST search, the glycoprotein hormone and relaxin receptors were identified as the most similar proteins. This shows that the resemblance of GRL101 to the other LGRs is not limited to the transmembrane regions and reinforces the hypothesis that the ligand of GRL101 has an insulin-like structure. GRL101 has a large number of LDLa’s, the ligands of which are typically positively charged surfaces, which in the case of proteins consist of Lys and Arg residues (Daly et al., 1995; Prévost & Raussens, 2004; Fisher, Beglova & Blacklow, 2006; Yasui, Nogi & Takagi, 2010; Dagil et al., 2013). Thus, the ligand of GRL101 may consist of two parts, an insulin-like structure and a piece with several positive charges that interact with the LDLa’s. The C-terminal tails of the IGFs, whether from arthropods, echinoderms or hemichordates, are all rich in charged amino acid residues. The C-terminal tail of IGF with its numerous positively charged amino acid residues might interact with the LDLa’s of GRL101. I, therefore, posit that in those species that have a GRL101 it functions as the second receptor for IGF. The absence of such a tail in GSS would make it likely that it acts on the RTK rather than an IGF GPCR.

The suggestion that GSS activates the RTK goes against the hypothesis that these peptides act through GPCRs. Indeed, it has recently been proposed that it is the ortholog of the dilp7 receptor that would be activated by the gonad stimulator in P. miniata (Mita et al., 2020). Given the clear orthology of both dilp7 echinoderm orthologs with the Drosophila peptide and the similar orthology between the dilp7 receptor and the echinoderm receptor, the conclusion that the two constitute a functional ligand receptor combination seems inescapable. It was impossible to find a GSS in either the genome assembly or the individual reads of all the genomic SRAs of Anneissia japonica, yet it does have a dilp7 receptor ortholog; thus, if the dilp7 receptor were to function as a GSS receptor, it most likely would not be an exclusive receptor. A priori, this does not exclude the possibility that GSS could function as a ligand for the same receptor. As mentioned above, since the dilp7 orthologs have well conserved F domains, one has to assume that this domain is important for binding to its receptor. Since the F domain is absent from RTK ligands, it is difficult to understand how a GSS that similarly lacks this domain would be able to bind the dilp7 receptor. It would, thus, seem unlikely that peptides as different as GSS and dilp7 would be effective ligands of the same LGR. Furthermore, the GSS genes have been duplicated, and their structures have diverged considerably. Those duplicate gonad stimulators are present in many species and have not been selected against. Hence they must be phyisologically relevant and able to interact with a receptor. Sharing a common evolutionary origin, the two gonad stimulators would be expected to act either on the same or paralogous receptors, but the number of putative echinoderm receptors for irps is limited, so they likely act on the same one. The same arguments that were used to argue that the multinsulins are likely RTK agonists but not LGR ligands, are therefore equally valid here and suggest that GSS is an RTK ligand.

Furhermore, the experimental evidence that GSS stimulates the ortholog of the dilp7 receptor is not convincing. The reported response to the dilp7 receptor when expressed in Sf9 cells is very weak and does not represent a typical response seen in this type of assay. Although the authors have shown high affinity binding of GSS to the follicle cells, such high affinity binding should also have been present in the Sf9 cells expressing the putative GSS receptor, but this was not reported. The follicle cell SRAs from which the putative GSS receptor was identified contain large amounts of RNAseq reads for the gonadulin receptor, a receptor that is more closely related to the vertebrate relaxin receptors than the dilp7 receptor, but surprisingly the authors do not mention this receptor, which they must have found (Mita et al., 2020).

I suggest that initially there was an IGF-like hormone that activated both a GPCR and an RTK. After two gene duplications some of the descendant ligands either lost their C-terminal tails or one acquired a larger one and this allowed all three ligands to activate, at least initially, the RTK while each acquired its own LGR. Later, some of the ligands may have lost their affinity for one receptor. Since the primary amino acid sequence of gonadulin is very different from that of the other irps, it likely lost its capacity to activate the RTK (Fig. 12). Holometabolous insect species have lost GRL101 and hence in those species IGF, can only act on the RTK. Under this hypothesis, the arginine-rich C-terminal tail would be useless in such insect species; in higher flies, such as Drosophila, it was indeed lost (Veenstra, 2020b). In vertebrates, there is no GRL101, and so IGF can only activate the two RTKs, while the relaxin related peptides are not known to interact with RTK. The presence of a similar arginine-rich E domain in the vertebrate IGF precursors might thus be an evolutionary relict.

Figure 12 How echinoderm irps may have evolved.

A represent an early metazoan in which an arch irp is a ligand for both an LGR and an RTK. B represents an early protostome or deuterostome that has three irps, an IGF and a dilp7 ortholog as well as gonadulin/octinsulin ortholog that evolved from local gene duplication from the arch irp. All three of these ligands each have their own LGR and at least two of them, IGF and the dilp7 ortholog, can also activate the RTK. C represents the Asterozoa where the dilp7 gene got duplicated and yielded several multinsulin genes which are represented here as one. The Asterozoa also have one or two GSS’s that evolved earlier during echinoderm evolution. Both multinsulins and GSS’s act exclusively through the RTK. Closed arrows indicate gene duplication events and interrupted arrows show ligand–receptor interactions. The question mark conveys uncertaintity with regard to whether or not the gonadulin/octinsulin peptides are able to activate the RTK.

This scheme raises the question as to how the functions of these two receptors activated by IGF might differ. IGF and the Drosophila irps stimulate growth, the echinoderm GSS stimulates oocyte maturation and ovulation (Mita et al., 2009), relaxin and INSL3 affet various developmental and reproductive processes (Ivell et al., 2020; Esteban-Lopez & Agoulnik, 2020), gonadulin is expressed by the gonads as well as the imaginal discs in flies (Garelli et al., 2012; Liao & Nässel, 2020; Veenstra, 2020b; Veenstra et al., 2021), and dilp7 is expressed in a sex specific manner (Miguel-Aliaga, Thor & Gould, 2008; Yang et al., 2008; Castellanos, Tang & Allan, 2013). These hormones stimulate growth, development and reproduction, processes that are intimately linked; without growth and development reproduction is impossible and growth without reproduction is useless in sexually reproducing species. On the other hand, resources used for growth and development can not be used for reproduction or vice versa.

Growth is rarely a linear process independent of development; animals are not only getting bigger, but they also mature into adults. Metamorphosis is markedly different between hemi- and holo-metabolous insect species. Every time a cockroach nymph molts, it becomes a little more adult, however during the first molts of a caterpillar the insects mainly become bigger, it is only when it molts into a pupa that it significantly changes its morphology. Cockroaches have GRL101; caterpillars don’t. This suggests that the RTK might be more directed toward linear growth, or allow growth by increasing uptake of resources, such as glucose and amino acids, while the LGRs might be more important for insuring that the animal develops into an adult and becomes sexually competent. Both holometabolous insects and vertebrates have lost GRL101 and use steroid hormones to induce sexual maturation. Interestingly, in vertebrates, the production of steroid hormones is controlled by glycoprotein hormones, the second group of ligands for LGRs.

It is plausible that IGF in an early bilaterian was produced by the tissue that stored energy and perhaps even protein as insects do in the form of storage proteins (Haunerland, 1996). Production and release of IGF might have happened when the animal had sufficient resources to allow for growth and/or reproduction. In arthropods, growth has become a discontinuous process in which a new cuticle needs to be made before molting can take place. In those species, IGF produced by the fat body may well be the essential growth hormone. However, if the animal is suddenly starved, IGF would no longer be released. If formation of a new cuticle is too advanced to be interrupted, this becomes problematic. It may have obliged the brain to take at least partial control of growth away from the fat body by releasing one or more of the neuroendocrine irps to force growth and molting to proceed. It is possible that this achieved by simultaneously reducing growth of organs that are needed for (sexual) maturation but not essential for immediate survival, like the gonads. This could be how the neuroendocrine insect irps initially evolved. In echinoderms, IGF probably stimulates growth of the follicles and oocytes, but the final growth spurt, the one that permits resumption of meiosis in the oocytes and subsequent ovulation, is delayed until optimal conditions to do so prevail. When the time and place are right, the nervous system releases GSS likely in large amounts to finish the maturation process and induce ovulation. In vertebrates, growth and the release of IGF has also been brought under control of the brain but more forcefully by bringing IGF secretion by the liver under control of growth hormone. Whereas in an early ancestor high plasma concentrations of glucose might have led to secretion of IGF, this is no longer the case. Here insulin may have evolved to insure that plasma concentrations of glucose are kept sufficiently low by insuring its absorption by tissues in order to avoid it loss by excretion. In the three cases these peptides have very different functions, ovulation in echinoderms, sparing glucose in vertebrates and rescuing interrupted growth in insects. It is plausible then that these hormones each evolved from a non-local IGF gene duplication and that they are thus not proper orthologs but evolved by convergent evolution. This hypothesis would explain, why there is no insulin gene located near the IGF, octinsulin/gonadulin and dilp7 triplet in cockroaches, echinoderms and hemichordates, even though insulin –and other peptides such as the insect neuroendocrine insulin-like peptides and GSS - almost certainly evolved from IGF much later.

Conclusions

The gene triplication previously reported from arthropods must have occurred in a common bilaterian ancestor of the deuterostomes and protostomes. The hypothesis that IGF in an ancestral bilaterian used both a GPCR and an RTK may explain the combination of echinoderm irps and putative insulin receptors. This hypothesis implies that insulin is not a hormone that evolved before the split between protostomes and deuterostomes, but that insulin-like peptides evolved independently in different metazoan clades as miniature copies of IGF capable to activate the RTK but unable to stimulate the LGR.

Supplemental Information

Supplemental Information 1 Supplemental Figures and SRA accession numbers

Click here for additional data file.

Supplemental Information 2 CDNA and protein sequences of ambulacrarian insulin related peptides and their putative receptors

Click here for additional data file.

Supplemental Information 3 Expression of ambulacrarian insulin related peptides and their putative receptors as determined from published transcriptome SRAs

Click here for additional data file.

This manuscript benefited from the critical contributions of an editor and two reviewers for which I am most grateful. Work like this is only possible because others made their transcriptome and genomic sequences publicly available. I express my sincere gratitude to all of them.

Additional Information and Declarations

Competing Interests

Author Contributions

Data Availability

The author declares that he has no competing interests.

Jan A. Veenstra conceived and designed the experiments, performed the experiments, analyzed the data, prepared figures and/or tables, authored or reviewed drafts of the paper, and approved the final draft.

The following information was supplied regarding data availability:

The SRAs are available at Genbank (File S1).

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
