# Peer review of "Ambulacrarian insulin-related peptides and their putative receptors suggest how insulin and similar peptides may have evolved from insulin-like growth factor"

_PeerJ, doi:10.7717/peerj.11799_

## Round 0.1 · original submission · Major Revisions

· Academic Editor

Major Revisions

Although both reviewers checked the "minor revisions" box, their recommendations are for somewhat more extensive revisions. Once revisions are made, the manuscript needs to be rereviewed. Reviewer no. 2, in particular noted that the nomenclature for the insulin-related growth factors is particularly confusing and needs to be clarified for the non-specialist. One example of such confusion is Fig. 9, which is a tree of the LGR-type receptors. The caption states that the inset at the top shows the same data to which the glycoprotein LGRs have been added and where characteristic ligands for each branch have been identified. This is totally confusing as the inset shows the ligands and the tree is not identical to the larger tree but includes two extra large branches, labeled Bursicon and GPA2/BPB5. If the intent is to show which ligands are bound by which receptors, that could be indicated at the right of the larger tree. The additional figures suggested by reviewer 2 would be very useful. A table listing the the different names for what the trees indicate are homologous peptides and their receptors would also be useful for the non-specialist. I look forward to a suitably revised version of the manuscript.

·

Basic reporting

no comment

Experimental design

no comment

Validity of the findings

no comment

Additional comments

This paper reports of a very well done analysis on the evolution of insulin-related peptides in ambulacraria. The paper is well written, the scientific context well introduced and the results are clearly presented and discussed.

I have only a few questions and suggestions that could help non expert readers to better understand the study:

The author uses for good reason "sequence similarity trees" approach analysis. The rationale for that is reported in the Materials and methods section where a very brief explanation of the approach is also reported. To help the non expert reader, I would give some general explanation in the Results section instead, and may be give more technical details in the Methods. For instance, it is stated that: "and Fasttree (Price, Dehal & Arkin, 2010) to construct trees and estimate probabilities", but no further details are given.

Similarly, there is quite a discussion in the sub sections "Nomenclature", "Sequence Analysis" and "Expression" of the Methods section, part of which would be more useful in the results, especially to orient the non expert reader who is only familiar with terms such insulin, IGF, etc.

Two other technical points are not clear to me and should be better specified:

1. where the genomic information for the following species was taken from?
- Anneissia japonica
- Holothuria glaberrima
- Patiria miniata
- Ophiothrix spiculata
I would assume from Echinobase, but I did not find this reference. Have I have missed it?

2. in the file "peerj-60199-Suppl.Spreadsheet1" there are several genes with Genbank identifier: NEW in red; what does it mean? are those sequences for which an accession number will be provider after publication?

·

Basic reporting

The peptide family described is quite complicated and the members have confusing names that make them easily mixed up. The terms "related" and "like" are not easily distinguishable. This makes it difficult to describe the situation clearly for non-experts. See my comments below. This is by no means the authors "fault", this is the way the peptides' names have gradually been assigned, making them quite challenging to describe. An introductory figure or table should help clarify the relationships. Likewise, a summarizing figure at the end can help describe the evolutionary scenario that the author proposes. I expect such clarifications would make the description easier to grasp for those who are not experts on this peptide and receptor family. I do not demand to see these clarifying additions, hence I have ticke "minor revision" rather than major, but I am prepared to do so if requested.

Experimental design

No comment.

Validity of the findings

No comment.

Additional comments

The terminology for this peptide family is quite confusing for non-experts (as are several other peptide families). It is difficult to distinguish the peptides that are described as IGF-like/related versus those that are called insulin-like, especially as the latter are considered to have emerged from the former by gene duplication. On line 65, this entire family of peptides is called "insulin-related peptides (irps)". So the family of insulin-related peptides started with an IGF-like that subsequently gave rise to insulin-like, then the "neuroendocrine insect irps" arose from their IGF-like. So it seems like irp is both the overarching name and a subset of IGF-like called neuroendocrine irps. Is it possible to describe this with a figure in the introduction? The author writes about the naming confusion on lines 99-108, especially regarding the receptors.

Another thing that would be very nice to have is a duplication scheme that summerizes the scenario that the authors favours.

24: It is confusing to add that the first of the three hormones is IGF-like when it has just been said that all three are IGF-related. Then on line 26 it is said that the peptides are insulin-like. One step in the evolution is missing, namely how two peptides went form being IGF-like to become insulin-like. Could it be that "Arthropod IGF" is the oldest and was duplicated, and this duplicate shrunk and became "insulin-like", and then evolved preference for a GPCR, and then this insulin-like gene was duplicated to give the third member. The two latter are what we call dilp7 and Arthropod gonadulin. Indeed, on line 88 the author writes about "two successive local gene duplications". The likely order of events gradually becomes clear as one reads the manuscript, but it is necessary that the abstract can summarize this in a reasonably intelligible way.

25: Why suggest that the ancestor of the three peptides used both types of receptors? Isn't it equally possible that one of the duplicates evolved preference for a second, and different, type of receptor? dilp7 is said on lines 76-77 to act on an RTK, in addition to its GPCR, so this peptide could represent the transition stage from one to the other and not necessarily an ancestral state of dual receptor preference. Again, the reasoning becauses clearer on reading the entire manuscript, but if this shall be presented in the abstract, it should ideally be well motivated here.

39-40 The author writes "If IGF were to act through both a GPCR and an RTK it would suggest that GSS acts on only one…". I am afraid I do not follow the reasoning here. If IGF worked on both, and gave rise to GSS, it would be expected that GSS too would initially act on both types of receptors, wouldn't it? But referring to my comment above, couldn't it be that IGF worked only through an RTK already before it gave rise to GSS, then GSS too would act on the RTK.

Minor comments

In a couple of places the author writes "genes coding peptides" or similary. It should "encoding peptides" or "coding for peptides", as indeed written correctly on line 263. Please scan the manuscript for this.

61 It is advisable to delete "closely" as some ligands bind to receptor subtypes that arose by duplication some 500 million years ago or more and have diverged considerably. I suggest that it is emphasized that this discussion concerns peptide ligands by inserting the word "peptide" before ligand(s) in this paragraph.

66 Correct the spelling of "coupled"

72 The sentence says "One of these" which seems to refer to dilp7 in the preceding sentence, but it should be "One of the irps".

86-87 The part after the comma is confusing. It does not necessarily have anything to do with the fact that the IGF gene is part of a local triplet.

112 Should be: number of Amulacraria species.

128 For those unfamiliar with these species, please write: "transcriptome data from the three crinoid species Antedon…."

154 should be "orthologous"

178 shold be "cleave any secreted protein with appropriate cleavage site".

206-211 This peptide organization would be good to illustrate with a figure.

216 "other echinoderms" should be "other ambulacrarians"

223 should be "cleavage sites"

258 it looks grammatically odd to see "are single copy", I suggest "is a single copy gene", this cannot be misunderstood.

260 Why write genus name for one and species name for the other?

308 should be "shows"
310 should be "where"

316-317: This sounds like a circular argument. Of course the tree based on sequence alignment and similarity will reflect the way they "look". Please rephrase.

324: should be NCBI

335: should be "multiple"

338-9: should be "from time to time"

339: "assembled constructs" is probably not the correct term here, I believe "assembled sequeces" would be more suitable.

340: "both" makes this sentence a bit difficult, please rephrase.

343: should be and adverb: "particularly"

352: "subbranches" should be followed by a comma.

357: "it is interesting to see" comments on something that cannot be seen in this clade in Fig. 9. Please describe differently or refer to a supplementary tree that contains this information.

363: What is "The second branch"? Please describe in an unambiguous way.

371: "are" should be "is"

372: "it are the glycoprotein hormone and relaxin receptors that are identified" can be corrected to "the glycoprotein hormone and relaxin receptors are identified"

373: add comma after GRL101.

386: it is more common to say "echinoderms"

391: "This" has unclear refrence and the sentence is dangerously close to circular reasoning: "pressure is weak…because it is poorly maintained" - please rephrase.

394: should be "suggest"

427: "with" should be "by"

430: As the sentence says "a second insulin RTK gene", it will have to be: "none was found".

431: "neither" does not fit here, better to write "were unsuccessful"

435-436: why can't binding to a second receptor have arisen later in evolution? See my comments at the beginning of my referee report.

463: "it were the…that were identified" can be simplified to: "the…receptors were identified…"

492: "evolved" can be replaced by "diverged"

Fig. 6. It would be very useful to have lines showing the organization of the prepro-peptides in relation to these schematic mRNA/cDNA outlines, and not only the Cys positions. Probably is would suffice with the prepropeptide outlines for the top and bottom sequences.

Fig. S11: Is TM1 correctly marked? I am not totally familiar with LRR TM regions but I think it might be shifted right-wards. All six sequences have an R in the TM1 region and four of them have a K a few positions further upstream. This would be a bit unusual to have within a TM region, but the LRR:s are a bit unusual so it may be correct. Please double check the boundaries.

---

## Round 0.2 · Minor Revisions

· Academic Editor

Minor Revisions

Although both reviewers felt that you had satisfactorily revised the manuscript, as neither one is a native English speaker, they made no suggestions for improvement of the writing. However, I found several instances where the punctuation or grammar made the manuscript hard to read. English pronouns are particularly difficult. For example, in line 45, the word "it" would, according to the rules, have to refer to "novel receptor." That makes no sense. Therefore, I suggested omitting the phrase "it seems to suggest." There were also a few misspellings of species names. I have, therefore uploaded a copy of the manuscript with some suggested changes. I look forward to receiving a suitably revised version of this interesting manuscript.
Sincerely,
Linda Holland

·

Basic reporting

N/A

Experimental design

N/A

Validity of the findings

N/A

Additional comments

I am fully satisfied of the explanation/corrections proposed by the author. In my former review I didn't raise a problem of nomenclature for the insulin-related growth factors, because I didn't have a solution to propose, but I see that, thanks to the suggestion of the other reviewer, the author found a solution which makes the manuscript more easily readable by both specialist and not specialist readers.
Overall I think that the manuscript improved considerably and represent a significant contribution to the field.

·

Basic reporting

No comment.

Experimental design

No comment.

Validity of the findings

No comment.

Additional comments

I feel that the renaming of the peptide groups now should make reading easier.

357: add missing 'o' in evolutionary

Fig. 1
This figure should help many readers. I just wonder if the D-domain has to start at the last residue of insulin, a single N. In insulin, this position is considered to be part of the A-chain. It would be difficult to argue that a single N constitutes a domain. But maybe there is a good reason to draw the boundary here that I am unaware of.

Previous Fig. 6, new Fig. 7
I had asked for prepropeptide outline to be displayed or marked, and the author wrote that this had been done, but the figure still looks the same to me. This is not a demand from me, I just thought it would be helpful.

---

## Round 0.3 · accepted · Accept

· Academic Editor

Accept

Dear Dr. Veenstra,

Thank you very much for submitting this thorough analysis of ambulacrarian insulin-like peptides to PeerJ.

I am glad that my copy editing was useful. It means the years I spent being taught English grammar and spelling were not entirely wasted.

The suggestion to send a tweet announcing acceptance is not mine but is automatically added by the journal.

Best Regards,

Linda Holland